# The Use of Thyme and Lemongrass Essential Oils in Cereal Technology—Effect on Wheat Dough Behavior and Bread Properties

**Lucie Jurkaninová** [1,*], **Ivan Švec** [2], **Iva Kučerová** [3], **Michaela Havrlentová** [4,5], **Matěj Božik** [1], **Pavel Klouček** [1] and **Olga Leuner** [3]

1 Faculty of Agrobiology, Food and Natural Resources, Czech University of Life Sciences Prague, Kamýcká 129, 165 00 Prague, Czech Republic; bozik@af.czu.cz (M.B.); kloucek@af.czu.cz (P.K.)

2 Faculty of Food and Biochemical Technology, University of Chemistry and Technology Prague, 166 28 Prague, Czech Republic; ivan.svec@vscht.cz

3 Faculty of Tropical AgriSciences, Czech University of Life Sciences Prague, Kamýcká 129, 165 00 Prague, Czech Republic; kucerovai@ftz.czu.cz (I.K.); leuner@ftz.czu.cz (O.L.)

4 Faculty of Natural Sciences, University of St. Cyril and Methodius in Trnava, 917 01 Trnava, Slovakia; michaela.havrlentova@ucm.sk

5 National Agricultural and Food Centre, Research Institute of Plant Production in Piešťany, 921 68 Piešťany, Slovakia

* Correspondence: jurkaninova@af.czu.cz

**Abstract:** Consumers are more interested in replacing commonly used chemical preservatives with natural substances. The effect of 5, 10, 20, 40, and 80 mg of thyme and lemongrass essential oils (THY and LMG, respectively) per 100 g of wheat flour was studied from the viewpoints of dough rheology, dough leavening progress, and the results of laboratory baking trial. Changes in dough viscoelastic properties were evaluated by the Mixolab rheometer and the company software. The higher the thyme oil portion, the higher the dough structure destruction by kneading and heat input in torque point C2, and consecutively shorter stability of dough consistency (up to one-half of the values recorded for the control); reversely, the LMG did not affect both features verifiably. In the 90 min leavening test, a dough weight loss was decelerated by both essential oils similarly. During the baking test, the average volume of wheat small breads as control was evaluated on level 167 mL (bread yield 451 mL/100 g flour). Independently of the dose of the THY or LMG, small bread volumes oscillated between 148–168 and 135–161 mL (average bread yields 442 and 443 mL/100 g flour, respectively). The shelf life of the products with a higher portion of essential oil was extended by up to 7 days.

**Keywords:** wheat flour; thyme essential oil; lemongrass essential oil; dough rheology; Mixolab; leavening test; bread volume; bread yield; bread shelf life

## 1. Introduction

Bakery products have become an important part of the daily diet of people around the world due to rapid socio-economic development [1]. More than 9 billion kilograms of bakery goods are produced each year and approximately 70 kg of bakery products are consumed per person per year [2]. Bread comes in many types, sizes, shapes, and textures depending on national and regional traditions [3]. At the same time, the consumption of bakery products made from wholegrain and graham flour is increasing gradually, mainly due to a greater awareness of the need to reduce the consumption of simple carbohydrates and fats. The consumption of wholegrain and graham flour products elevates the intake of dietary fiber and vegetable protein [4].

To produce bakery products, the basic ingredients are flour, water, yeast (or other fluffing-up agent), and salt. Additional raw materials can be used to improve dough processing or to manufacture special products that often have a higher nutritional value [5]. The optimal processing depends on the type of baked goods and the requested properties of

the final product [6]. For baked goods, important quality characteristics include large loaf volume, soft, uniform texture, and satisfactory shelf life [2]. Rolls and breads have a limited shelf life. They undergo physical, chemical, and microbial changes during cooling after baking and storage itself. Physical and chemical changes cause the loss of freshness, and deterioration in texture and taste. Microbiological spoilage is caused due to the growth of bacteria (mainly *Bacillus subtilis*), fungi (especially *Rhizopus nigricans*, *Penicillium expansum* or *P. stolonifer*, and *Aspergillus niger*), and yeasts (most often *Saccharomices cerevisiae*) [3]. It is responsible for the change in appearance, odor formation, and the production of toxic metabolites, monitored are mainly aflatoxin, ochratoxin A, deoxynivalenol, and zearalenone. These can threaten human health and generate significant economic losses for the baking industry and consumers [7]. According to the Food and Agriculture Organization of the United Nations (FAO), foodborne molds and their toxic metabolites cause about 25% of the global agricultural food losses [8].

The spoilage of bakery products is influenced by several factors. The importance could be considered for processing conditions including packaging and product characteristics. For a product, the ingredients used, nutritional composition, oxidative stability, moisture, water activity, and pH are decisive. In processing, the baking time, baking temperature, cooling, and hygiene of the production environment matter. For storage, temperature, access to light, relative humidity, and microorganism content are controlled. Packaging characteristics are also very important. For packaging, mechanical properties, thermal stability, gas permeability, UV barrier, gas composition, antioxidant, and antimicrobial activity are monitored [1].

A set of procedures is used to prevent the spoilage of bakery products. Conventional methods use preservatives. Preservatives are added to foods to increase or maintain nutritional value and quality. They prolong the shelf life of the products reasonably. They inhibit the growth of microorganisms and thus support consumer acceptability [9]. The amount and composition of each preservative is regulated by legislation. Preservatives are divided into two groups, chemical and natural ones. Chemical preservatives include synthetic substances such as benzoates, sorbates, nitrites, nitrates, sulfites, glutamates, glycerides, and others [10]. Natural preservatives include compounds that are obtained from natural sources such as salt, sugar, vinegar, honey, spices, and other substances [9].

Consumers have become more interested in healthy lifestyle forms and diets in recent years. There is a growing desire to reduce commonly used chemical preservatives (e.g., 'non-E' brand). In this context, essential oils are one of the important replacements for their antioxidant, antibacterial, antifungal, antiviral, antiparasitic, and insecticidal properties [11]. Moreover, the aroma of essential oils can positively enhance the sensory properties and increase the attractiveness of baked goods.

Essential oils are complex mixtures of the volatile compounds produced by plants. They can be synthesized in all plant organs, flower, bud, seed, leaf, stem, fruit, and root [12]. They are the secondary metabolites of aromatic plants [13]. Essential oils provide them protection against bacteria, viruses, fungi, and insects, and also enhance plant protection against herbivores [14].

Essential oil consists of approximately 20–60 different components. Usually, two or three components are dominant from 20–70%, while the remaining are present in trace amounts only [15]. The major components are mainly responsible for the biological function of the essential oil, but the remaining components also play a partial role [8].

Differences in the chemical composition and biological activity of essential oils depend on the climatic conditions of the growing area, the cultivation strategy, the plant variety, fertilization, and sufficient water intake during the growing season of the plant [16]. Variations in composition may also be due to the part of the plant from which they are obtained [15].

Spices and essential oils are used in food processing to enhance the flavor and extend the shelf life of foods [17]. The antimicrobials of plant origin led to an increase in overall quality and to the removal or reduction in pathogenic microorganisms. For example,

eugenol in clove, cinnamaldehyde in cinnamon, carvacrol and thymol in oregano, sulfur compounds mainly diallyl trisulfide and diallyl disulfide in garlic, linalool in coriander, 1,8-cineol and camphor in rosemary, pulegone and limonene in parsley, citral in lemongrass, camphor in sage, and vanillin in vanilla have shown great antimicrobial effects [18].

However, the application of essential oils has some limitations despite its great potential. Essential oils can cause an allergic reaction and, from a certain amount, cause acute and chronic toxicity. Their use is also limited by the lack of scientific documentation for correct exposure dosages [14]. Another problem is their chemical variability. As the secondary metabolites of plants, essential oils are affected by external factors. This makes them extremely variable qualitatively and quantitatively over time. They have significant technological limitations for their application. The challenges are quality and standardization and strong sensory impact. Essential oils are volatile, unstable in air, and prone to oxidation. The extremely lipophilic nature of essential oils makes them difficult to use in polar solvents [19].

Thyme (*Thymus vulgaris* L.) is an aromatic, culinary, and medicinal herb of the *Lamiaceae* family. It is native to the Mediterranean region [11]. The main components of thyme essential oil are thymol and carvacrol, in levels of up to 70%, and p-Cymene in levels of up to 40%. The ratio of the components varies according to the species, chemotype, or geographical origin. The remainder consists of linalool, limonene, geraniol, borneol, terpinene, and camphene [20]. The high antioxidant activity is mainly due to flavonoids, but phenolic compounds such as rosmarinic and caffeic acids, thymol, and carvacrol also exhibit antioxidant activity [21]. Lemongrass (*Cymbopogon winterianus* Jowitt or *Cymbopogon nardus* (L.) W. Watson) is a grass of Southeast Asia. It contains a significant amount of essential oil consisting mainly of citronellal, geraniol, and citral, which is characterized by a lemony aroma due to the citral compound. The essential oil exhibits medicinal, antifungal, antimicrobial, and antioxidant effects [22].

The main reasons for using essential oils in bakery are their antimicrobial and antioxidant effects. To increase shelf-life, essential oils can be added as a direct ingredient in the formulation of bakery products or can be added to the packaging material or the internal atmosphere of the package [10]. Bakery products with medium and high water activity are sensitive to microbiological spoilage. These products include filled bakery products, soft biscuits, and laminated bakery products. Almost all bacteria, yeasts, and molds can occur in bakery products with a water activity above 0.94. Breads, fruit pies, cakes, and pizzas fall into this category [23]. The antibacterial effects of essential oils are manifested by limiting bacterial growth or directly destroying bacterial cells [8]. In general, Gram-positive bacterial species are more sensitive to natural compounds than Gram-negative bacteria due to the absence of an outer membrane [24]. The antibacterial activity of different essential oils has been investigated against both Gram-positive bacteria (*Bacillus subtilis*, *Staphylococcus aureus*, and *Listeria monocytogenes*) and Gram-negative bacteria (*Escherichia coli*, *Salmonella typhimurium*, *Pseudomonas aeruginosa*, and *Camplyobacter* spp.) [8]. Both cinnamon and clove essential oils can inhibit both Gram-positive and Gram-negative bacteria [24]. Mustard, cinnamon, and clove essential oils slow down the spoilage process of baked goods. Lemon grass, cinnamon essential oil, clove essential oil, and thyme essential oil have fungicidal activity. Turmeric, lemongrass, and cloves when added to the recipe retarded the growth of mold in butter cakes [25]. Bakery products are prone to rancidity [25]. Antioxidant activity may also be associated with non-phenolic compounds such as limonene, linalool, and citral. Cinnamon and clove essential oils consist of various monoterpenes and sesquiterpenes that show significant antioxidant activity; especially, cinnamaldehyde and eugenol are associated with antioxidant effects [8].

Rheological measurements can simulate and characterize the material properties during processing, thus allowing the quality control of products [26]. The rheological properties of dough are good indicators of behavior during kneading, fermentation, and baking [27]. Mixolab is a modern instrument for determining the quality control of cereals and flour. It is used to determine the thermomechanical properties of dough. The advantage

of this method is that operating conditions such as kneader speed and temperature can be determined. This makes it possible to assess the effect of the addition of additives, kneading parameters, and the properties of the dough during heating [28]. For the analysis, 75 g of dough is used to determine the quality of protein and starch. The standard 'Chopin+' protocol analyzes dough kneading behavior, starch lubrication, amylase activity, and starch retrogradation in a single test [29]. Essential oils can influence the physicochemical and rheological properties of the dough. There are changes in the qualitative properties of baked goods such as volume, texture, and sensory characteristics [27]. The volume of the bread usually decreases, and the stiffness increases after the addition of essential oils. Essential oils are thought to bind to storage proteins or polysaccharides depending on their nature. This leads to a strengthening of the dough and a prolongation of the dough development time [30]. The use of thyme essential oil at concentrations equal to or greater than 0.15 mL/100 g of dough resulted in a significant reduction in bread volume [31].

The present study was aimed at the assessment of the effect of thyme and lemongrass essential oils on the bakery quality of common commercial wheat flour in several technological stages. Changes in behavior during kneading, i.e., in the viscoelastic properties of non-fermented dough were recorded on the Mixolab instrument. The course of fermentation was described by a row of the standard leavening microtests, and the final quality of bread in terms of laboratory baking trial was finalized by color evaluation.

## 2. Materials and Methods

### 2.1. Materials

Based on information from the literature and previous research, we determined the effective concentrations of essential oils (EOs) as follows: 0 mg/100 g for the control, and 5, 10, 20, 40, and 80 mg/100 g of flour. To ensure accurate EO concentration and to simplify dosing, the amounts were recalculated according to the actual density of both essential oils, and application was performed by a micropipette.

The wheat white, finely-milled flour (STD) was rendered by the producer Mlýny J. Voženílek, Private Limited Company, settled in town Předměřice nad Labem, Czech Republic. According to the Czech state standard (ČSN 56 0512 [32], corresponds to the ICC 104/1 [33]), the quality parameters specify an ash content of a maximum of 0.52% and a gluten content in dry matter of a minimum of 32.0%.

Thyme essential oil (THY) as well as lemongrass essential oil (LMG) were bought from Sigma-Aldrich Ltd. (Saint Louis, MO, USA), with the countries of origin being Spain and India, respectively, produced by the steam distillation method, thyme EO (44.34% thymol, 17.88% p-Cymene), lemongrass EO (40.06% geranial, 31.65% neral), determined by GC-MS and GC-FID.

The density of the THY was 0.917 g/mL, and of the LMG 0.896 g/mL. Both EOs were mixed with the STD immediately before each test, and selected doses increased stepwise from 0 mg/100 g of flour to 5, 10, 20, 40, and 80 mg/100 g of flour. The codes of all flour mixtures combined the actual dose and the type of EO: 5THY, 10THY, 20THY, 40THY, 80THY for the THY, and similarly 5LMG–80LMG for LMG.

For the leavening microtests, further raw materials were necessary: fresh pressed baker's yeast (*Saccharomyces cerevisiae*) 'EXTRA' was produced by the LLC 'BALEX COMPANY' (Kharkov, Ukraine), while salt and beet sugar were bought in the local retail market. In correspondence with the work's aim of EO effect testing, any other commonly used fat (rapeseed or sunflower oil, or margarine) was not used.

### 2.2. Methods

#### 2.2.1. Mixolab Test

To describe the rheological behavior of wheat flour enhanced by essential oils, rheometer Mixolab (Chopin Technologies, Villeneuve-la-Garenne, France; today part of KMP Analytics, Westborough, MA, USA) with the company's software was employed. An advantage of this apparatus lies in low material consumption as well as in the description

of the dough kneading phase together with the dough pasting one during one test (in principle, it represents a combination of the Farinograph rheometer and the Amylograph viscometer). The test settings included the 'Chopin+' protocol according to the ICC norm No. 173; details of the protocol settings were published earlier [34,35]. The description of the curves recorded was restricted to the primary parameters: torque points C1, C2, C3, C4, and C5 and times $t$_C1 as well as STA (time of the occurrence of the torque point C1 ≈ dough development time, and stability of dough consistency). The technological meaning of these five torque points was discussed in the same papers [34,35].

For each single test with the essential oil, a determination of the moisture content was carried out in advance by using an infrared moisture balance. Wheat flour mixture with THY or LMG was prepared manually—10 g of finely milled wheat flour was weighed into the beaker and the desired amount of essential oil was added with a pipette. The flour and essential oil were mixed carefully with a glass rod, and homogenization of the sample took 2 min. The mass was poured onto an aluminum plate and placed in an infrared balance. For the Mixolab test itself, flour–oil samples were prepared similarly; only a mixture homogenization was prolonged to 5 min.

### 2.2.2. Leavening Microtest—Dough Volume & Sample Weight Monitoring

A leavening (fermentation) test was carried out according to the internal methodology. It is the simplest method to monitor volume changes in yeasted dough during a technological phase of leavening. For more detailed and precise records, apparatuses called fermentograph or rheofermentometer could be employed after yeasted dough preparation on a laboratory kneader (typically the Farinograph or the AlveoConsistograph). Here, the recipe comprised 50.0 g flour, 1.5 g fresh compressed yeast, and 40 mL distilled water (30 °C). The correct amount of essential oil was pipetted into the flour, followed by the thorough homogenization of the sample by manual mixing for 5 min. A water suspension was prepared from the yeast first, followed by the preparation of the dough. Yeast suspension was combined with the flour–EO mixture and kneaded for 3 min to a uniform structure. The prepared dough mass was quantitatively transferred into 250 mL glass beakers or into 50 mL Falcon tubes for weight and volume monitoring, respectively. Each fulfilled laboratory glass was placed in a fermentation tank where the leavening test was carried out under pre-set fermentation conditions (34 °C, relative humidity (RH) 78%). The fermentation experiment took 90 min in total, and both the weight and volume of the samples were recorded at 10 min intervals. The weight was evaluated with an accuracy of ±0.001 g. Each result is the average of two measurements.

The leavening test was used also for testing lupin flour effect on semolina breadmaking quality—Spina et al. [36] prepared dough on Farinograph to consistency 500 Brabender units, weighted pieces of 25 g and measured in 50 mL gradual cylinders, previously oiled to avoid sticking. For dough leavening, they set the incubator to 30 ± 2 °C, and contrary to our own procedure, measurement in each 10 min was aimed at reading the volume level only.

### 2.2.3. Bread Preparation and Quality Evaluation

For the laboratory baking experiment, the internal modification of the standard Rapid Mix Test (RMT) method was used. The baking trial aimed at quality control and the evaluation of the effect of EO addition to the recipe on bread quality, bread volume, and sensory acceptability (ECC 2062/81). The used recipe is as follows: 200 g finely-milled white wheat flour (T530), 3.3 g salt, 114 mL water, 6.0 g baker's fresh yeast; THY or LMG in doses 5, 10, 20, 40, and 80 mg/100 g of flour. As in the case of the leavening test, the recipe did not involve any other fat.

In a Domino food processor (výrobce, město, stát) equipped with a kneading hook, the homogenization of the flour–EO mixtures took 2 min, followed by fast 2 min mixing with further ingredients and final slow 10 min kneading of dough. In an incubator with auto-controlled temperature and RH pre-set to 34 °C and 78% (výrobce, město, stát), dough mass

and dough pieces spent 20 and 40 min for leavening and maturation. After the leavening stage, the dough mass was split into $60.0 \pm 0.1$ g pieces manually and rounded up on the Extensograph shaping unit. Baking was carried out in a professional oven (výrobce, město, stát)–(*nebo* spolupracující pekárna); the program 'BUN' with auto-steaming and an initial temperature of 260 °C, gradually reduced to 237 °C in 13 min was selected. Baked pieces were left on filtration paper for 2 h to cool down at laboratory conditions. Three representative samples were weighted, and their volumes were determined by the traditional rapeseed displacement method (AACCI Method 10-05.01) [37]. The collected data were transformed into bread yield on the basis of the known proportion of flour into 1 small-bread piece.

Bread yield was calculated by Equation (1):

$$Bread\ yield = \frac{100 \times volume\ of\ 3\ bulks}{weight\ of\ flour\ in\ 3\ bulks}\ \ (\text{mL}/100\ \text{gflour}) \tag{1}$$

### 2.2.4. Bread Color Assessment

The evaluation of the color of the bread parts (crumb and crust separately) was made by using a colorimeter CM-600d (Minolta, Osaka, Japan) at a 10° angle and in daylight mode (D65). The CIE *Lab* colorimetric model was adopted, measuring the color coordinates $L^*$ (luminosity, whiteness), $a^*$ (greenness to redness), and $b^*$ (blueness to yellowness). The measurements were carried out in five repetitions using two standards, crust standard $L^*$ 67.5, $a^*$ 11.47, and $b^*$ 36.42, and crumb standard $L^*$ 75.06, $a^*$ 0.17, and $b^*$ 19.03. Before each measurement, the instrument was calibrated using White Calibration Cap CM-A177.

In the CIE *Lab* color space, the difference between two colors (tints) is expressed by Equation (2):

$$\Delta E = \sqrt{(\Delta L^*)^2 + (\Delta a^*)^2 + (\Delta b^*}\ )^2 \tag{2}$$

The final $\Delta E$ value is the difference of the measured pairs ($L^*_{\text{SAMPLE}} - L^*_{\text{STD}}$), ($a^*_{\text{SAMPLE}} - a^*_{\text{STD}}$), and ($b^*_{\text{SAMPLE}} - b^*_{\text{STD}}$) [38]. Simplified, it is a rectangular (Euclidean) distance between the tint of the enriched product and the tint of the control (e.g., bread, or crust and crumb separately). There is experimental knowledge about the so-called *just noticeable difference* between two independent samples—by human vision, they could be considered as clearly different (recognizable) at $\Delta E$ equal to 1.0–2.0 at least.

### 2.2.5. Statistical Analysis

Data collected from the rheological measurement, leavening microtest, and baking trial was described by Tukey's HSD test and correlation analysis ($p = 95\%$) using the Statistica 13.0 software (TIBCO Software Inc., Palo Alto, CA, USA). The interrelations among the qualitative features were explored by multivariate Principal Component Analysis (PCA). The procedure of the PCA was carried out in two steps—firstly with the complete dataset to identify the most representative features, whose variability was explained by the first three principal components (PC) from 60% at least. At the same time, the representative features must cover all specific technological phases of bread manufacturing as dough kneading, leavening test, baking trial, and bread color assessment. For example, information doubling was confirmed between the Mixolab features including the stability of dough consistency and the torque point C2 as heat–mechanical dough destruction ($r = 0.95$, $p = 95\%$), and reversely, an irreplaceable role could be addressed to the dose of the essential oil and dough weight at the 90th min of the leavening test. After such reduction, PCA was repeated with an improved explanation rate of the scatter of the representative data.

## 3. Results

### *3.1. Mixolab Test*

The non-enriched wheat flour (STD) was characterized by a standard dough development time (parameter $t\_C1$) of 3.75 min and long stability of consistency (equal to 9.20 min;

Table 1). As supposed, the addition of both essential oils did not change values of the specific torque points C1 and C3–C5 verifiably, perhaps owing to low dosages and the absence of alpha-amylases, acting in the second pasting phase of the Mixolab test. Only in the case of dough mechanical–thermal destruction and the proper torque point C2, a somewhat higher extent of the gluten skeleton damage was observed for the dough with THY doses 20, 40, and 80 mg/100 g flour. Such a difference could be later reflected in bread volume.

**Table 1.** Mixolab testing of rheological behavior of non-fermented dough as affected by several doses of thyme and lemongrass essential oils (THY and LMG, respectively).

| Essential Oil | | Mixolab Torque Point | | | | | Mixolab Time Value | |
|---|---|---|---|---|---|---|---|---|
| **Type** | **Dose (mg/100 g)** | **C1 (N.m)** | **C2 (N.m)** | **C3 (N.m)** | **C4 (N.m)** | **C5 (N.m)** | ***t*_C1 * (min)** | **STA ** (min)** |
| Control | 0 | 1.08 a | 0.50 a | 1.82 a | 1.68 a | 2.61 a | 3.75 a | 9.20 d |
| THY | 10 | 1.10 a | 0.51 a | 1.82 a | 1.67 a | 2.60 a | 3.95 ab | 8.10 c |
| | 20 | 1.12 a | 0.41 a | 1.79 a | 1.71 a | 2.82 a | 3.73 a | 5.90 a |
| | 40 | 1.12 a | 0.41 a | 1.82 a | 1.71 a | 2.80 a | 3.90 ab | 5.80 a |
| | 80 | 1.09 a | 0.41 a | 1.81 a | 1.71 a | 2.87 a | 3.97 ab | 6.90 b |
| LMG | 10 | 1.08 a | 0.51 a | 1.85 a | 1.67 a | 2.66 a | 3.70 a | 9.40 d |
| | 20 | 1.10 a | 0.52 a | 1.86 a | 1.74 a | 2.74 a | 3.93 ab | 9.20 d |
| | 40 | 1.07 a | 0.51 a | 1.83 a | 1.68 a | 2.65 a | 3.88 ab | 9.30 d |
| | 80 | 1.07 a | 0.53 a | 1.84 a | 1.63 a | 2.67 a | 4.18 b | 9.20 d |
| Subgroup means | | | | | | | | |
| Control | 0 | 1.08 A | 0.50 AB | 1.82 A | 1.68 A | 2.61 A | 3.75 A | 9.20 B |
| THY | 10–80 | 1.11 A | 0.43 A | 1.81 A | 1.70 A | 2.77 A | 3.89 A | 6.68 A |
| LMG | 10–80 | 1.08 A | 0.52 B | 1.84 A | 1.68 A | 2.68 A | 3.92 A | 9.28 B |

* time of the occurrence of the maximal consistency C1; ** stability of dough consistency. a–d: averages in columns, signed by the same letter, are not statistically different (*p* = 95%). A–B: In columns, capital letters mark subgroup averages and statistically significant differences (*p* = 95%).

Only the time parameters of the Mixolab test were varied significantly—more than the time of C1 point occurrence (data variance $a − b$ only), just the stability of dough consistency was shortened by THY to approximately a half (from 9.20 min to 6.68 ± 1.07 min; data variance $a − d$). The LMG counterpart had no recognizable influence on that dough cohesivity. Pooling over the EO doses, a distinguishing of THY-dough in the stability of consistency from the STD and LMG counterparts was statistically verified (*p* = 95%) (Table 1).

### 3.2. Leavening Test

In a brief overview, the leavening microtest demonstrated its ability to describe satisfyingly a course of wheat dough fermentation progress under step-by-step doubled additions of the two types of essential oil.

### 3.2.1. Dough Weight Loss

Both for dough weight and volume, data recorded within the leavening microtest could be processed by three-factorial ANOVA; the F1 was leavening test time, F2 essential oil type, and F3 essential oil dose. For the dough weight loss, a linear decrease for the non-fortified wheat dough (STD) as well as for both the THY- and LMG-enriched counterparts was observed. A weight loss step oscillated between 0.1 and 0.2 wt.%; to ensure the clarity of the ANOVA chart and to magnify significant differences, a dataset was reduced to the leavening times 0, 30, 60, and 90 min (Figure 1). During the entire leavening test, 1.21 g of the control wheat dough was metabolized by yeasts. For the 5THY and 10THY samples, the consumed mass portion increased nearly twice (1.97 and 2.04 g, respectively). Along with the magnifying THY doses, these total weight losses diminished surprisingly up to 1.34 g. For a row of the LMG-dough variants, the corresponding weight losses were lower—1.59 g,

1.76 g, and 1.38 g, respectively. Described differences in the effect of THY and LMG can be noticed in Figure 1, especially for specimen pairs 5THY–5LMG and 10THY–10LMG.

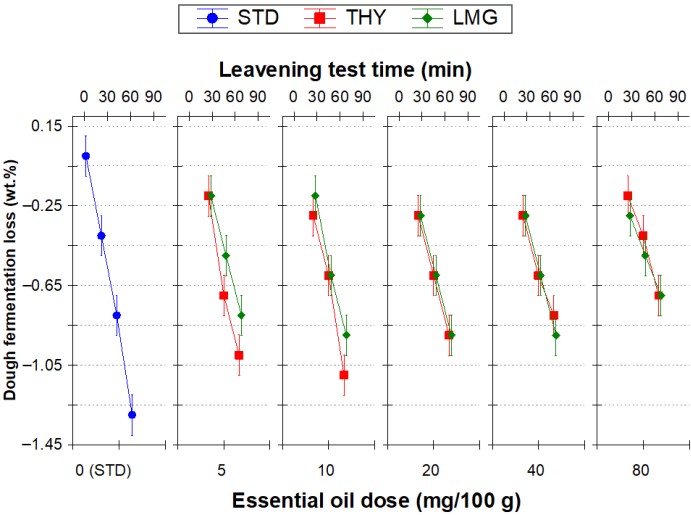

**Figure 1.** Comparison of the reduced results of the 90 min leavening microtest of the weight loss of the non-fortified wheat dough (STD) as influenced by the stepwise-rising doses of the thyme or lemongrass essential oil (THY and LMG). Vertical bars denote 0.95 confidence intervals.

### 3.2.2. Dough Volume Monitoring

In terms of dough volume monitoring, the effect of the THY and LMG was reversed—in general, the higher the oil content in the leavened dough, the harder the progress of the fermentation process. In bakeries, there is a common praxis to even double a recipe dose of yeasts in the case of the manufacturing of fat-rich Christmas or Eastern pastry. The maximal volume of the wheat control was 42 mL, which occurred at the 70th min of the leavening test (423% of the initial volume). The maxima of dough volumes of all 10 EO-enriched doughs did not level to the control (Figure 2); in the case of THY additions, observed values ranged from 31 to 38 mL (at 50th to 70th min of the leavening test; 308 and 383% of the initial volume). A stronger negative influence of the LMG oil is documented in the data pictured in Figure 2—dough volume maxima reached 29–36 mL.

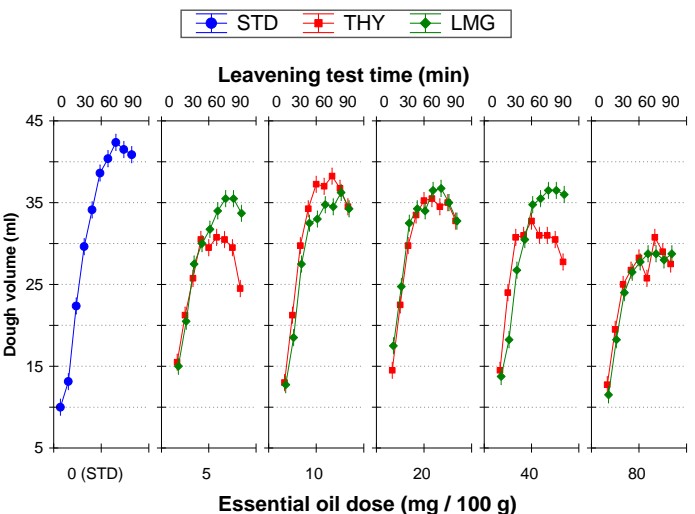

**Figure 2.** Comparison of the complete results of the 90 min leavening microtest of the volume rise of the non-fortified wheat dough (STD) as influenced by the stepwise-rising doses of the thyme or lemongrass essential oil (5, 10, 20, 40, and 80 mg/100 g of flour). Vertical bars denote 0.95 confidence intervals.

*3.3. Baking Trial*

3.3.1. Bread Yield

The control bread sample (STD) had a bread yield of 451 ± 16 mL/100 g flour. The bread yield of the samples with the addition of the THY essential oil ranged from 401 to 455 mL/100 g flour (Figure 3; average 440 ± 27 mL/100 g flour). However, data oscillation for these THY-bread variants did not demonstrate any trend (the lowest bread yield 401 mL/100 g flour found for 10THY, while the highest 478 mL/100 g flour for 20THY blend). Similarly, the additions of the LMG essential oil varied the bread volumes independently to the applied dose—the yield of the LMG-breads ranged from 405 to 482 mL/100 g flour (average 441 ± 25 mL/100 g flour; Figure 3). The bread from the 20LMG blend had the highest yield, and reversely the bread from blend 80LMG. In general, the antimicrobial activity of essential oils can have a toxic effect on yeasts or inhibit yeast activity and reduce fermentation speed. Fat may slow down the fermentation process via the greasing of starch granule surface and hardening of their accessibility to amylases.

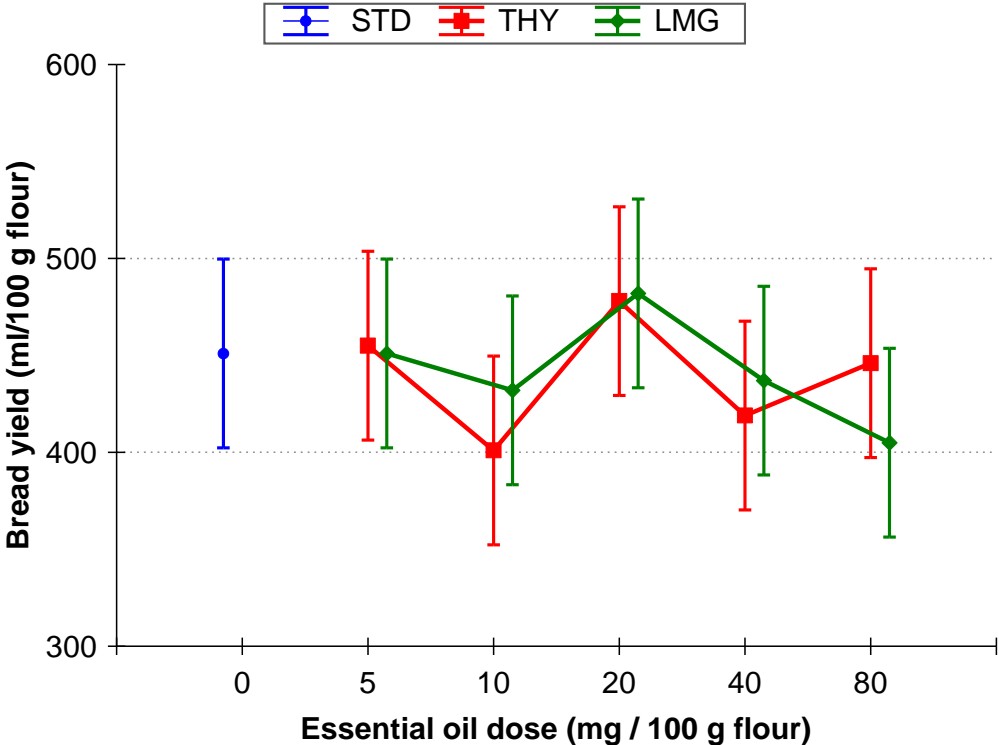

**Figure 3.** Comparison of the baking trial results of the non-fortified wheat bread (STD) as influenced by the stepwise-rising doses of the thyme or lemongrass essential oil (5, 10, 20, 40, and 80 mg/100 g of flour). Pictured differences were insignificant completely (two-factorial ANOVA, *p* = 95%). Vertical bars denote 0.95 confidence intervals.

To document changes in small-bread shape and crumb porosity, bread cuts were scanned on a common office scanner at a resolution of 300 dpi. In Figure 4, wheat control and variants enriched by 80 mg THY or LMG/100 g of flour are presented. As can be noticed, the higher doses of EO affected the bread height, which corresponds with non-fermented dough elasticity (extensograph or alveograph one). The higher diameter of the breads with 80 mg THY or LMG/100 g of flour thus reflects a supporting effect of EO on dough extensibility.

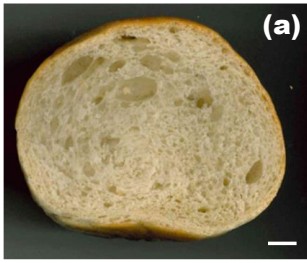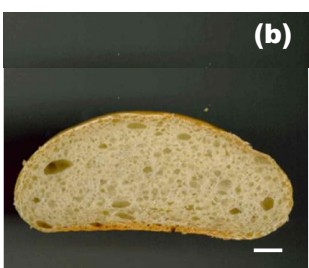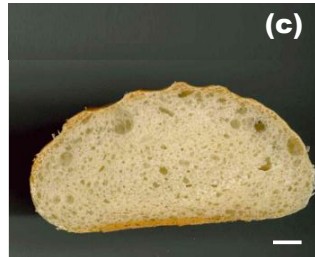

**Figure 4.** Comparison of cut appearance of selected small-bread variants. (**a**) unfortified wheat control, (**b**) bread with 80 mg thyme essential oil/100 g of flour, (**c**) bread with 80 mg lemongrass essential oil/100 g of flour. Note: scanned on common office scanner at resolution 300 dpi.

### 3.3.2. Bread Color Evaluation

The changes in the color of the bread crusts fortified by THY and LMG essential oils in increasing levels are summarized in Table 2a. By using a CM-600d spectrophotometer, the color of the crust of the control bread (STD) was characterized by lightness $L^*$ 67.5, and the values of redness $a^*$ 11.47 and yellowness $b^*$ 36.42.

**Table 2.** (**a**) Crust color of wheat bread variants tested as affected by several doses of thyme and lemongrass essential oils (THY and LMG, respectively); (**b**) Crumb color of wheat bread variants tested as affected by several doses of thyme and lemongrass essential oils (THY and LMG, respectively).

| (a) | | | | | |
|---|---|---|---|---|---|
| **Essential Oil (EO)** | | **Crust Whiteness *L*** | **Crust Redness *a*** | **Crust Yellowness *b*** | **Crust Δ*E*** |
| **Type** | **Dose (mg/100 g Flour)** | | | | |
| Control | 0 | 67.49 c | 11.47 abc | 36.42 ab | 0.00 a |
| THY | 5 | 64.74 abc | 12.10 abcd | 38.33 b | 5.16 bc |
| | 20 | 66.48 bc | 12.67 abcd | 34.79 ab | 3.35 bc |
| | 80 | 60.00 ab | 15.70 cd | 37.23 ab | 8.82 d |
| LMG | 5 | 69.83 c | 9.96 ab | 32.98 a | 4.01 bc |
| | 20 | 63.40 abc | 12.36 abcd | 34.50 ab | 3.05 b |
| | 80 | 67.17 c | 12.69 abcd | 36.15 ab | 4.15 bc |
| Subgroup means | | | | | |
| Control | 0 | 67.49 A | 11.47 A | 36.42 A | 0.00 A |
| THY | 5–80 | 63.70 A | 13.10 A | 35.94 A | 6.30 C |
| LMG | 5–80 | 66.00 A | 12.69 A | 36.70 A | 4.00 B |
| (b) | | | | | |
| **Essential Oil (EO)** | | **Crumb Whiteness *L*** | **Crumb Redness *a*** | **Crumb Yellowness *b*** | **Crumb Δ*E*** |
| **Type** | **Dose (mg/100 g Flour)** | | | | |
| Control | 0 | 75.06 cd | 0.17 a | 19.03 ab | 0.00 a |
| THY | 5 | 77.49 def | 0.18 a | 18.45 a | 2.31 bc |
| | 20 | 79.43 f | 0.27 abc | 19.39 ab | 4.39 d |
| | 80 | 78.39 ef | 0.35 abcd | 19.66 bc | 3.47 cd |
| LMG | 5 | 72.56 bc | 0.42 cd | 19.29 ab | 3.45 cd |
| | 20 | 66.87 a | 0.46 cd | 19.23 ab | 8.55 e |
| | 80 | 71.64 b | 0.50 d | 20.65 c | 3.81 cd |
| Subgroup means | | | | | |
| Control | 0 | 75.06 B | 0.17 A | 19.03 A | 0.00 A |
| THY | 5–80 | 78.00 B | 0.26 A | 19.11 A | 3.00 AB |
| LMG | 5–80 | 71.20 A | 0.42 B | 19.43 A | 4.20 B |

*L*—luminosity, whiteness, *a*—greenness to redness, *b*—blueness to yellowness; Δ*E*—total color difference; in the CIE *Lab* space, Euclidean (i.e., rectangular) distance between the color of the fortified and control bread variant. a–f: averages in columns, signed by the same letter, are not statistically different ($p$ = 95%). A–C: In columns, capital letters mark subgroup averages and statistically significant differences ($p$ = 95%).

In the case of the crust tint of the fortified bread variants, the observed *L\** values ranged between 59.0 and 69.8, while the *a\** and *b\** values were determined in ranges 8.79–15.70 and 32.98–38.33, respectively. There was no evidence of a trend in the change in bread crust color between both essential oil types tested, as well as among their doses in the recipe. Significant differences were observed between the 40THY and 5LMG samples, which showed significantly lower redness *a\** values and at the same time higher whiteness *L\** values. No higher extent of the darkening of the breads' crust was recognized along with the rising doses of both THY and LMG essential oils.

In the evaluation of bread crumb color in the CIE *Lab* space, the position of the control with any essential oil (STD) was [*L\**; *a\**; *b\**] = [75.1; 0.17; 19.03]. The crumb color coordinates of all 11 bread samples ranged in interval 66.9–79.4 for lightness *L\**, 0.17–0.50 for redness *a\**, and 18.45–20.65 for yellowness *b\**. Similar to the bread crust color measurement, no rising or diminishing trend was observed (Table 2b).

Both for crust and crumb color of single bread samples, any tendency connected to essential oil type or essential oil dose was not proven also for the total color difference Δ*E* (overlaying ranges 3.1–9.0 for the crust color, 0.7–8.6 for the crumb color). Considering an average color change as a result of THY or LMG incorporation, the tint of the crust was distinguished completely and the one of the crumb partially (Table 2a and Table 2b, respectively). As mentioned above, an average appearance of crust control and LMG-breads could be considered as just noticeably different to each other; moreover, bread counterparts with THY oil could be recognized both from the control as well as from ones containing LMG oil. In the case of the bread crumb tint, only the pairs of control—THY-breads and control—LMG-breads met the condition of a minimum difference in Δ*E* equal to 2.0 (Figure 5).

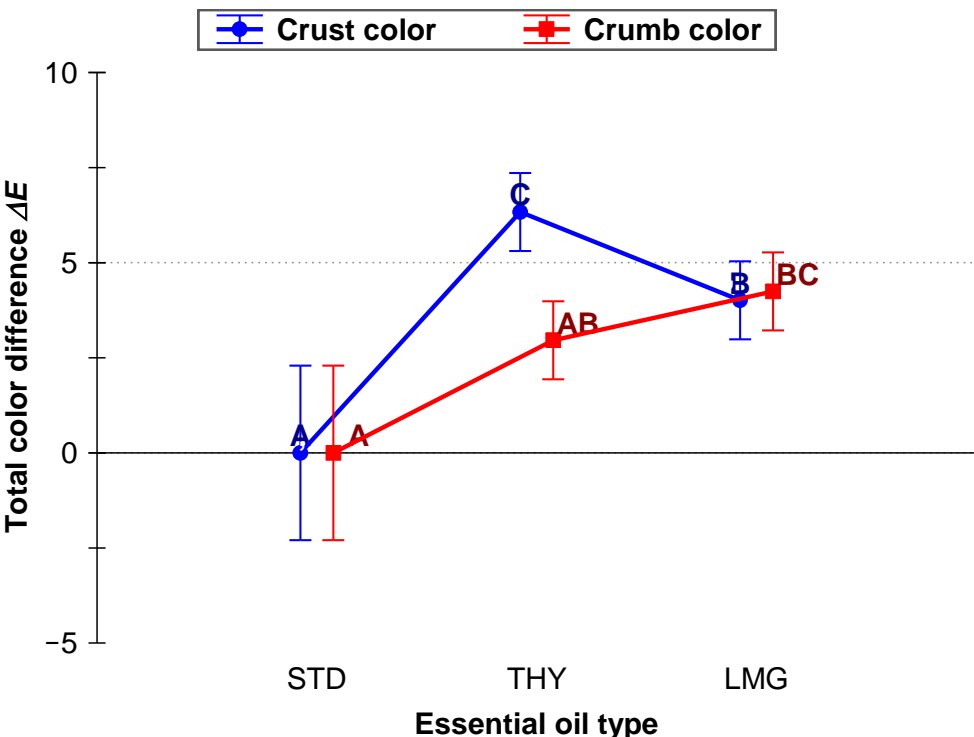

**Figure 5.** Comparison of the average total color difference Δ*E* between the crust and crumb of the non-fortified bread (STD) and its counterparts enriched by thyme and lemongrass essential oils (THY and LMG, respectively). A–C: averages signed by the same letter are not statistically different (*p* = 95%). Note: average values pooled over all five doses of essential oil applied (5–80 mg/100 g flour). Vertical bars denote 0.95 confidence intervals.

### 3.4. Multivariate Statistical Analysis

After the original dataset reduction, mentioned in Section 2.2 above, 9 variables from the original 19 ones were maintained as the necessary as well as sufficient ones for a description of the recorded data variability. In correspondence with information compressed in Table 3, the first principal component (PC1) covered 46%, the PC2 22%, and the PC3 10% of the reduced dataset scatter on average. The PC1 covered mainly the Mixolab torque points of dough consistency and hot gel stability (C1 and C4) plus the crumb whiteness $L*$; the PC2 is built-up from the characteristics essential oil dose, starch retrogradation rate (torque point C5), and dough weight loss in 90 min of the leavening test (Figure 6a). For the majority of the mentioned dough quality features, these PC1 and PC2 could be considered sufficient to ensure a minimal information loss (communalities sum $\geq$ 70%). For the bread yield additionally, the PC3 should also be included—as mentioned above, bread volume variance did not allow a clear distinguishing of essential oil type, or its dose used. It is a partially logical finding that the volume and shape of the final leavened bakery product depend on many factors; they begin from used dough recipe and mixer type via leavening conditions and dough pieces shaping procedure up to baking conditions. The plot of PC1 × PC2 loadings confirmed a known positive relationship between specific bread volume/bread yield and the Mixolab torque point C2—the more resistant the dough to heat–mechanical treatment, the potentially larger rise of the dough during leaving (based on stronger gluten net present in that dough).

**Table 3.** Communalities (%), a percentage of explained data variability by the first three principal components (PC).

| Variable | | PC1 | PC2 | PC3 | Sum |
|---|---|---|---|---|---|
| Dose of essential oil | | 3 | 45 | 29 | 77 |
| Mixolab torque point | C1 | 74 | 3 | 6 | 82 |
| | C2 | 64 | 24 | 3 | 91 |
| | C4 | 41 | 0 | 1 | 42 |
| | C5 | 41 | 47 | 3 | 91 |
| Baking trial | Dough weight$_{90\,min}$ | 14 | 53 | 6 | 73 |
| | Bread yield | 27 | 2 | 42 | 70 |
| Bread color | Crumb $L*$ | 86 | 1 | 0 | 87 |
| | Crumb $a*$ | 64 | 25 | 0 | 89 |
| Average | | 46 | 22 | 10 | 78 |

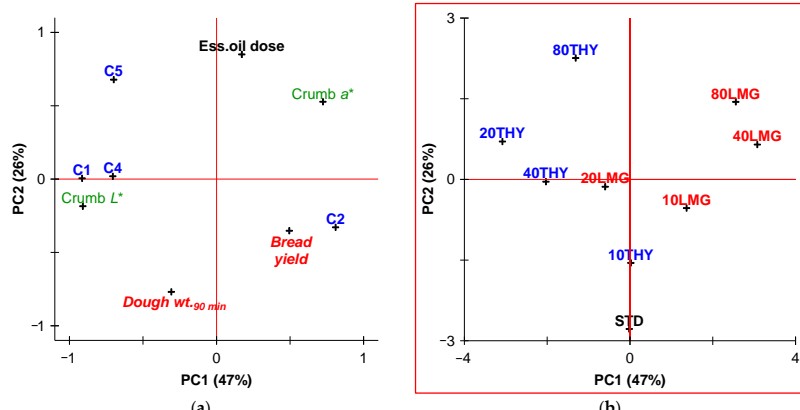

**Figure 6.** Principal component plot: (**a**) of variable loadings, where C1, C2, C4, and C5: Mixolab torque points; Dough wt.$_{90\,min}$—weight of the dough sample in the 90 min of the leavening test; *Crumb L\**, *Crumb a\**—crumb luminosity (whiteness) and redness; (**b**) of case scores, where STD—non-fortified wheat dough/bread; 10-20-40-80THY/LMG—wheat dough/bread fortified by 10 up to 80 mg/100 g four of thyme and lemongrass essential oil, respectively.

Correspondingly to the positions of the variables in Figure 6a, the tested samples are located in the same plane of the PC1 × PC2. The effects of THY and LMG on flour/dough and bread properties are listed above; in the plot Figure 6b, there is documented a statistical closeness of the samples 10THY and 10LMG to the non-enriched control (STD).

Uncovered oscillation mainly in bread yields and crumb color caused shifts in the position of the samples containing 20 or 40 mg/100 g of both essential oils (20THY, 40THY, 20LMG, and 40LMG), whilst as presumed, the most enriched specimen 80THY and 80LMG are the furthest from the control (STD).

## 4. Discussion

### 4.1. Mixolab Test

In the previous case of the Czech commercial flour testing on the Mixolab, the time of reaching the C1 torque point was prolonged by about 4.72 to a value of 8.47 min. Similarly, the dough stability as resistance to over-kneading was supported in the same direction (11.53 min) [39]. Durum wheat is known through a different proportion of gliadins and glutenins, and thus a diverse rheological behavior of non-fermented dough. In Northern Italy in the year 2014, common and durum wheat reached a shorter dough development time (1.8 and 1.3 min) than both the Czech samples, but different stability (10.0 and 3.2 min, respectively) [40]. The addition of 1 wt.% and 3 wt.% of THY into wheat flour magnified the Farinograph dough development time of wheat flour from 1.9 min to 8.3 and 15.9 min, respectively. An insignificant effect was evaluated for the same doses of wheat germ oil (from 1.9 min to 2.0 and 1.8 min) [30]. The authors also tested further essential oils from the seeds of black currant and black cumin, and the Farinograph test results ended between the mentioned ones recorded for the wheat germ and THY. The dough stability of their wheat flour was 8.3 min, and the listed plant oils varied that value both negatively and positively (5.3–15.5 min). For bakers, more important knowledge was gained in terms of dough softening, which did not overcome the value of 40 Brabender units (BU) for the control wheat flour. For the industrial production of the Central European types of leavened small-bread as rolls and buns, the recommended Farinograph dough development time oscillates around 3.0 min with dough softening degrees up to 100 BU.

### 4.2. Leavening Test

The leavening test does not belong among the preferred proofs of wheat flour/dough quality and owing to that, there is a lack of reference materials. In research aimed at bakery applications, moreover, essential oils are tested mainly in the aspect of their effect on a bread's quality as well as the antifungal agent. Debonne et al. [31] found a notable decrease in bread volume when thyme essential oil was added. They attributed this reduction to the impact of thyme oil on the yeast viability in the dough, which affects the optimal speed of bread volume rise. In paper [36], a leavening test was used to describe the course of semolina dough leavening under several additions of sweet lupin flour and lupin protein concentrate (3–15 wt.%). Both lupine materials increased slightly Farinograph water absorption, but significantly prolonged the dough development time and reversely the dough stability. In the leavening test, consequently, more elastic semolina–lupin dough underwent a fermentation process more quickly, reaching maximal volume in approximately half the time in contrast to the control semolina dough. These effects may be addressed to either the higher fat content in lupine flour (6–8%) or to a partial disruption of too cohesive semolina dough itself. In contrast to lupine flour, white or black chia seeds are richer in fat content (30–35%) [41]. During the dough preparation of the Farinograph, wheat flour replaced by 5 or 10 wt.% of both chia flour variants needed about 4.8 % pt. and 9.6 % pt. more water to reach the prescribed consistency of 600 units. By using the machine Fermentograph SJA, leavening proof taking 160 min brought knowledge about the one-tenth shorter optimal leavening time (time of the first reaching of the maximal dough volume). At the end of this 160 min proof, dough volumes were lessened to approx. 71% and 56% without the effect of chia seed type (seed color). Regardless of the lower portions

of fermentation gases captured and maintained in the wheat–chia dough, total production of the fermentation gases was almost comparable among all five samples (687 mL for control wheat dough, and 630–615 mL vs. 645–650 mL for wheat–white chia/black chia dough variants, respectively) [42].

### 4.3. Bread Yield and Color

An important parameter indicating the quality of the baked product is its final volume, which is related to the entire technological process, but mainly to the amount and quality of protein, especially gluten [43]. The addition of THY and LMG essential oils to the recipe did not lead to a significant reduction in volume yield compared to the control. The concentrations tested ranged from 0 to 80 mg/100 g of flour. Debonne et al. [31] reported a negative effect of thyme essential oil on bread volume from a concentration of 137.55 mg/100 g flour (172% of own maximal dose). Lower concentrations had no damaging effect on the specific volume of the bakery product. According to Dhillon et al. [44], dried thyme positively impacted bread quality when supplemented even up to 2% of flour. This was evident in the increased specific volume values at 1% and 2% supplementation levels. On the other side, spices and herbs such as cinnamon, clove, garlic, oregano, and thyme, added at 1 or 2%, significantly decreased the bread volume [45]. Lemongrass essential oil is well known for its strong antimicrobial activity against fungi, yeasts, and bacteria [46]. There is a presumption of yeast inhibition and as a result of the reduction in fermentation, the volume of bread may be reduced [31].

The observed $L^*$ values for the crust were measured from 59.0 to 69.8, which is a standard range determined by many other authors [47]. The values for $a^*$ ranged from 8.79 to 15.70 and $b^*$ from 32.98 to 38.33 indicating the crust color of elaborated breads fell within the red–yellow area, which is a favorable characteristic of baked goods for consumers. Johnston et al. [48] used white wheat all-purpose flour, produced in New Zealand, and tested wheat bread fortification by wholemeal faba bean flour and protein isolate. Their control wheat bread had similar $L^*a^*b^*$ color coordinates ([59.9; 13.1; 31.3] for crust and [70.22; 2.07; 19.18] for crumb) as the own control (STD). The addition of 20 wt.% of both faba bean forms changed the color of wheat bread crust significantly on all three axes of the color space—the evaluated coordinates were [45.5; 16.5; 29.5] and [31.2; 15.0; 17.9], respectively. The authors explained bread browning, reddening, and yellowing as just a result of the Maillard reaction, speeded up by a higher content of legume proteins.

The color of the crust and crumb are quality parameters, which are associated with the organoleptic properties of the bread [49], and these attributes hold significance from the perspective of the customers. Caramelization and Maillard reaction led to the development of non-enzymatic browning in baked goods surface (resulting in its attractive flavor and beige–brown color). Reducing sugars react with an amine, creating via glycosylamine a derivate of amino deoxy-fructose. Our results indicate that the addition of EO to the recipe did not manifest itself in a large and consumer-unfriendly way. For example, the addition of protein-rich plant materials, e.g., chickpea powder, supports crust browning. Replacing 5.0, 17.5, and 30.0 wt.% of wheat flour, the $\Delta E$ has risen from 24.9 and 30.2 up to 34.1, respectively [50]. The total color difference $\Delta E$ can be classified analytically according to Cserhalmi et al. [51] as not noticeable (0–0.5), slightly noticeable (0.5–1.5), noticeable (1.5–3.0), well visible (3.0–6.0) and great (>6.0). Thus, $\Delta E$ of the crust is categorized as well visible and, only in the case of THY doses 10 and 80 mg/100 g flour, is considered as great. However, evaluating this total color difference of crumb is more variable and is categorized as noticeable (40THY and 10LMG), great (20LMG), and well visible for the rest. Debonne et al. [31] reported the discoloration of bread after the incorporation of thyme essential oil into a recipe as a result of the oxidation process that the essential oil undergoes, causing changes in food color. Thus, it is evident that applying EO can influence the final color of the bread.

In the realm of food science research, one of the primary challenges confronting enriched bakery food production is the harmonization of fortification with optimal sen-

sory attributes [52]. Together with product volume, color is one of the primary sensory attributes influencing consumers' behavior [53] and is strongly associated with the concept of quality [54]. The secondary sensory attributes are crust crispiness and crumb resilience (elasticity).

## 5. Conclusions

The aim of the study was to evaluate the impact of the use of selected essential oils on the viscoelastic properties of wheat dough, the fermentation process course, and the final quality of yeasted bakery products. In lower concentrations, the inhibition of leavening is not dramatic and does not have a negative effect on the technological process and the final quality of the small-piece bread. The results of the Mixolab rheological tests showed a possible influence of the essential oils used on the weakening of the gluten structure, which became significant during the second heating–cooling phase of the proof, potentially affecting bread volume. No effect on the water absorption, kneading, or development of the dough was detected. The comprehensive examination of rheological behavior, leavening processes, and baking outcomes provides valuable insights into the potential applications of thyme and lemongrass essential oils in bakery formulations. The clarification of the effects of essential oils on dough rheology, fermentation processes, and fermentation kinetics provides essential insights for optimizing dough performance and final product quality.

In the baking trials, both thyme and lemongrass essential oils did not significantly reduce the bread volume yield compared to the wheat control. Although the color of crust and crumb was influenced by essential oil type and dose, variations observed were maintained within acceptable ranges for consumers' preference. This finding is particularly significant for the food industry, as it suggests that essential oil enrichment can be integrated into bakery formulations without compromising product volume or quality. For all nine manufactured bread variants, an indicative sensory analysis was appended. At higher concentrations (40 or 80 mg/100 g flour), the impact on the bread taste was enormous. From the point of view of consumer's acceptability, the concentration of essential oils was too high, manifesting flavor of categories 'excessively intense' or even 'unacceptable', accompanied by feelings of bitterness, pungency, and burning with a long-lasting effect in the mouth. However, the lower concentrations were very interesting and potentially attractive to common consumers.

It is important to find the optimal level between the effective dose of the specific essential oil, consumers' preference, and formulation as well as bakery technology process optimization and the suitable method of essential oil application. In industrial bakeries, for example, ready-to-use flour-based blends are commonly processed. If the essential oils could become a part of these premixes, there is a challenge for the premix producer to consider protection against potential oxidation.

Overall, the study provided a partial insight into the complex interactions between essential oils and dough characteristics, offering opportunities for optimizing bakery processes and product quality. Given the huge potential of natural substances, including essential oils, it is necessary to continue research and find the best ways to use natural substances.

It is necessary to test different doses of thyme and lemongrass essential oils and find the ideal concentration to achieve the desired dough properties and quality of the final product. In baking recipes, it is important to strike a balance between the characteristic flavor, antioxidation effect, and potential health benefits of essential oils and functional considerations such as dough consistency and leavening, as well as bread volume. The increased knowledge of the interaction of thyme and lemongrass essential oils with yeast activity and dough fermentation can inform adjustments in proofing time, yeast dosage, and processing conditions to optimize product proofing and consistency.

Bakeries can take advantage of market trends and focus on developing innovative product lines that take advantage of the unique sensory and health benefits of essential oils to meet consumer demand for healthier and more flavorful products. Informing consumers about the use of essential oils in bakery products can increase transparency and build

confidence about the safety and potential health benefits of essential oils. This educational approach can help consumers make informed choices and promote a positive perception of the bakery branch.

**Author Contributions:** Methodology, L.J., I.Š. and I.K.; Software, I.Š.; Formal analysis, L.J., M.B. and P.K.; Investigation, L.J., I.Š., I.K. and P.K.; Writing—original draft, L.J. and I.Š.; Writing—review & editing, M.H. and O.L. All authors have read and agreed to the published version of the manuscript.

**Funding:** This work was supported by the National Agency for Agricultural Research of the Ministry of Agriculture of the Czech Republic under project Biostore (QK21010064). The work used [data/tools/services/facilities] provided by the METROFOOD-CZ Research Infrastructure (https://metrofood.cz; accessed on 30 May 2024), supported by the Ministry of Education, Youth, and Sports of the Czech Republic (Project No. LM2023064).

**Institutional Review Board Statement:** Not applicable.

**Informed Consent Statement:** Not applicable.

**Data Availability Statement:** The data presented in this study are available on request from the corresponding author.

**Conflicts of Interest:** The authors declare no conflicts of interest.

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
