# Peer review of "The Use of Thyme and Lemongrass Essential Oils in Cereal Technology—Effect on Wheat Dough Behavior and Bread Properties"

_applsci, doi:10.3390/app14114831_

Round 1

Reviewer 1 Report

Comments and Suggestions for Authors

In this work, the effect of thyme and lemongrass essential oils on bakery quality of common commercial wheat flour in several technological stages was studied from viewpoints of dough rheology, dough leavening progress, and results of laboratory baking trial. Changes in behavior during kneading was recorded on the Mixolab instrument. Course of fermentation was described by a row of the standard leavening microtests, and the final quality of bread, in terms of laboratory baking trial, finalized by color evaluation. They found that the shelf life of products with a higher portion of essential oil was extended by up to 7 days and suggested that essential oil enrichment can be integrated into bakery formulations without compromising product volume or quality. They concluded that comprehensive examination of rheological behavior, fermentation processes and baking results provided valuable information on the potential applications of thyme and lemongrass essential oils in bakery formulations.

I have an observation about the manuscript:
Place tables and graphs in appropriate places to ensure good reading of the manuscript... e.g.
...move Figure 1b after section 3.2.2. Dough Volume Monitoring (pages 338-348)
...move Table 2a after section 3.4.2. Bread color evaluation (pages 372-375)
...move Table 2b after section 3.4.2. Bread color evaluation (pages 392-396)
...move Figure 3 after section 3.4.2. Bread color evaluation (pages 406-406)

Reviewer 2 Report

Comments and Suggestions for Authors

Subject: Manuscript revision entitled "The use of thyme and lemongrass essential oils in cereal technology – effect on wheat dough behavior and bread properties"

 In the study “The use of thyme and lemongrass essential oils in cereal technology – effect on wheat dough behavior and bread properties”, the effect of thyme and lemongrass essential oils on bakery quality of common commercial wheat flour in several technological stages was investigated. The manuscript addresses a topic of great interest nowadays, given the growing demand for bioactive compounds with preservative properties, however, there will some minor revisions and modifications that need to be performed.

Comments are included in the manuscript.

Author Response

Dear reviewer,

thank you for your expert review, your time and attention to our work.

We appreciate your evaluation and suggestions for improving the paper. Thank you.

We have placed a more detailed recap in the PDF document as a response to your comments.

Best regards.

Lucie Jurkaninová and author's collective

Reviewer 3 Report

Comments and Suggestions for Authors

The effects of two essential oils on the characteristics of wheat dough and bread were investigated, which have potential application value.

(1)The figures in the manuscript needs to be modified. I suggest changing Figure 1a and Figure 1b to Figure 1 and Figure 2.

(2)The position of Figure 2 in the text needs to be checked. I did not see the illustration of Figure 2.

(3)The color parameters of the standard board need to be given.

(4)I suggest that the author can provide a picture of the appearance of the bread, which is very attractive to the reader.
